# Extrinsic and Intrinsic Factors Determine Expression Levels of Gap Junction-Forming Connexins in the Mammalian Retina

**DOI:** 10.3390/biom13071119

**Published:** 2023-07-13

**Authors:** Tamás Kovács-Öller, Gergely Szarka, Gyula Hoffmann, Loretta Péntek, Gréta Valentin, Liliana Ross, Béla Völgyi

**Affiliations:** 1Szentágothai Research Centre, University of Pécs, 7624 Pécs, Hungary; kovacs-oller.tamas@pte.hu (T.K.-Ö.); jerjely@gamma.ttk.pte.hu (G.S.); hgyula@gamma.ttk.pte.hu (G.H.); pentekloretta00@gmail.com (L.P.); g.vtin83@gmail.com (G.V.); 2Department of Neurobiology, University of Pécs, 7624 Pécs, Hungary; 3NEURON-066 Rethealthsi Research Group, 7624 Pécs, Hungary; 4Center for Neuroscience, University of Pécs, 7624 Pécs, Hungary; 5Faculty of Science, University of Calgary, Calgary, AB T2N 1N4, Canada; liliana.ross@ucalgary.ca

**Keywords:** retina, gap junction, connexin, circadian rhythm, development, light adaptation

## Abstract

Gap junctions (GJs) are not static bridges; instead, GJs as well as the molecular building block connexin (Cx) proteins undergo major expression changes in the degenerating retinal tissue. Various progressive diseases, including retinitis pigmentosa, glaucoma, age-related retinal degeneration, etc., affect neurons of the retina and thus their neuronal connections endure irreversible changes as well. Although Cx expression changes might be the hallmarks of tissue deterioration, GJs are not static bridges and as such they undergo adaptive changes even in healthy tissue to respond to the ever-changing environment. It is, therefore, imperative to determine these latter adaptive changes in GJ functionality as well as in their morphology and Cx makeup to identify and distinguish them from alterations following tissue deterioration. In this review, we summarize GJ alterations that take place in healthy retinal tissue and occur on three different time scales: throughout the entire lifespan, during daily changes and as a result of quick changes of light adaptation.

## 1. Introduction

The mammalian retina is a sheet of nervous tissue that covers the posterior aspect of the eye. Its neurons are organized into three cellular layers incorporating the light-sensitive photoreceptors (PRs), three interneuron populations including bipolar cells (BCs) horizontal cells (HCs) and amacrine cells (ACs), as well as the retinal ganglion cells (RGCs) that provide the sole output of the retina towards the brain. Neurons in the retina like in other brain areas communicate via chemical and/or electrical synapses. Following the initial descriptions [1,2,3,4], electrical synapses (also called gap junctions—GJ) were considered scarce and were thought to play insignificant roles, but their essential interplay in neuronal signaling became evident just recently. Experimental work over the past 3 decades also showed that GJs are just as important in the retina as in any other brain area [5,6]. Although the retinal distribution pattern of the retinal connexins (Cx; molecular building blocks of GJs) gained firm support, it is well documented that their expression levels are subject to changes in the external and internal milieu. The sections of this review will summarize some of these changes.

### 1.1. GJs and Connexin Building Blocks

GJs serve as conduits allowing for the free transcellular diffusion of ions and small molecules up to 1 kD [1,2], which paves the way for a molecular exchange between cells. In addition to small molecules, charged ions pass through GJs as well, thereby altering the membrane potential of both cells, which is a process that is utilized by neurons to perform fast and two-directional signaling. Due to this latter phenomenon, GJs are also called electrical synapses. At GJ sites, the membranes of neighboring cells form close appositions leaving only a very thin (1–2 nm) synaptic space in between. These intimate physical contacts harbor a piece of molecular machinery whose integral parts are the so-called connexons/hemichannels on each side. A functional channel is formed by two opposing hemichannels that are located in plasma membranes of neighboring cells. Each connexon is formed by a hexameric conglomerate of six membrane-spanning connexin (Cx) protein subunits. Cx-s are formed by four transmembrane domains: two extracellular and one intracellular loops and cytoplasmic C- and N-terminal endings. The amino acid sequence and the Cx makeup determine the molecular permeability and unitary conductance of each pore. The functional properties of the pore can also be modified by the intracellular molecular milieu, and thus the on- and offset of various signal-transduction pathways [7]. In the human and mouse genomes that have been studied most extensively, over 20 Cx genes were identified and sequenced [8]. Cx proteins are named simply after their molecular weights expressed in kilodaltons (kDa; ranging between 21–70 kDa, including Cx43, Cx40, Cx26, etc.). The Cx phylogenetic tree has three main branches comprising α-connexins (e.g., Cx38 and Cx40, Cx43, Cx45, Cx46, Cx50) with long intracellular C- and N-terminal chains, small β-connexins (e.g., Cx26, Cx30.2, Cx31 and Cx32) with short intracellular N-terminal [7] and medium-sized γ-connexins (e.g., mammalian Cx36, skate, perch and zebrafish Cx35, perch Cx34.7). In contrast to α- and/or β-Cx-s, pores formed by γ-Cx-s have unique features as they have low (or no) voltage- and pH-sensitivity and are also unable to form heterologous channels with α- and β-Cx subunits. It has now been firmly established that Cx-s are important in development, differentiation and growth control in all organs, and every vertebrate class uses a unique set of Cx-s to build GJs [9].

### 1.2. GJs in the Retina

GJs are abundant in all brain areas including the mammalian retina (Figure 1), where all major neuron classes have been shown to form GJs to couple neighbor cells [6,10,11,12,13,14]. In addition to the great variety of connecting cell types, the diverse Cx makeup also suggests that retinal GJs play a number of roles in signal processing. The best-studied GJ sites in the vertebrate retina are those highly conductive GJs that couple HCs into extensive laterally oriented syncytia. GJs of this circuit are used to average signals of the ambient background light across the coupled HC array, the signal of which can be then utilized by BCs to detect contrasts prevailing against this uniform background. Homologous GJ synapses (connecting neurons of the same subtype) are commonly used to average noise, thereby increasing the signal-to-noise ratio. Such noise-reducing GJs exist in both plexiform layers, including the cone–cone GJs in the outer retina as well as those that couple AII ACs in extended homologous arrays in the inner retina [15]. On the other hand, heterologous GJs connecting rods to cone PRs provide an alternative route for rod-mediated signals [6,16,17,18,19,20,21]. Finally, a cohort of studies showed that GJs formed between RGC neighbors or connecting RGCs to nearby ACs partake in the correlation of the RGC spike output sent towards the brain [22,23,24,25,26]. Overall, these data indicate that electrical synaptic circuitry in the retina is as complex and dynamic as the well-described circuits formed by chemical neurotransmitters.

### 1.3. Connexin Composition of Retinal GJs

Cx36 has been found in both plexiform layers [27] of the retina of various mammalian species, whereas other subunits are restricted to either plexiform layer (Figure 2). Cx45 for example can be found mostly in the inner retina [28], whereas both Cx50 and Cx57 are expressed by the outer retinal HCs and thus their distribution is confined to the OPL [29,30]. Cx36 GJs have been reported in multiple sites of the rod pathways, including AII AC junctions and those formed between rod and cone PRs [27,31,32,33,34,35,36]. In the proximal retina, Cx45 has been localized to some of the axon terminals of ON cone BCs at sites where they form GJs with the Cx36-expressing AII ACs. While these latter connections are clearly heterotypic, the rest of the AII AC/ON cone BC connections are homotypic and show the presence of Cx36 in both neuron populations [37,38]. Therefore, both Cx36 and Cx45 are essential subunits for transmitting rod-mediated signals for night vision pathways. Apart from mediating signals of the night vision pathways, both Cx36 and Cx45 participate in RGC circuits to serve visual feature encoding. It has been shown for instance that Cx36 is expressed by α-GCs and that Cx45 comprises GJs of ON–OFF direction-selective RGCs, serving spike synchronization and encoding of the direction of movement, respectively [39,40,41,42]. In addition to Cx36 and Cx45, only one other subunit, Cx30.2, has been shown to be expressed by A1 type of RGCs [43]. Besides the retinal distribution pattern of the Cx protein subunits, the expression levels of corresponding Cx36 mRNA transcripts have been studied as well, and their distribution patterns correlated with those of the subunit proteins [44,45,46]. However, GJs are not static bridges but, similar to chemical synapses, their prevalence and gating properties are highly regulated by changes throughout the postnatal development by external factors like adaptation to various light conditions and/or by the circadian master clock [38,47]. Many of the rapid changes in GJ coupling appear to be due to changes in post-translational modification (i.e., phosphorylation) rather than changes in protein expression, but longer lasting intrinsic and/or environmental alterations also induce expression changes as well. The following sections will point out the milestones in the corresponding field of research.

## 2. Postnatal Changes in Connexin Expression

Over 50 years ago, Potter and his colleagues [48] provided electrophysiological evidence that the squid embryo maintained low-resistance conduits between certain cells during development that are subsequently lost in the adult animal. Soon after this first demonstration, it became clear that this phenomenon is not unique to the squid embryo, but it is rather ubiquitous in the animal kingdom. GJs have been found in both invertebrates including the sea urchin, starfish, ascidians (note that invertebrates utilize innexin proteins to build GJs) and various species from all vertebrate classes [49]. Since those early days, it has repeatedly been shown that the expression of GJ-forming Cx transcripts display a temporal pattern during organogenesis both outside the brain [50,51,52] and in many brain areas [53,54,55,56,57]. Cx36 expression in the murine retina for example displayed an increased expression from P1 to adult retinal stages in the two plexiform layers [58]. Quantitative polymerase chain reaction (qPCR) and western blot (WB) measurements in the rat retina also showed a gradual increase in Cx36 expression in both the mRNA transcript and protein levels [46], suggesting that the described developmental expression changes are ubiquitous among mammalian species. It appears that Cx36 immunoreactive puncta appear first in a few cells in the presumptive ganglion cell layer at P1 [45] and become abundant in the two plexiform layers at P10 [46]. This latter developmental stage also coincides with a peculiar surge in Cx36 protein expression in WB measurements and just precedes the maximum of Cx36 mRNA expression at P15 [45,46]. The Cx36 expression peak also overlaps with the observable colocalizations between Cx36 puncta and parvalbumin (PV)-expressing rat AII amacrine cell dendrites in strata 3 and 4, whereas those located on AII amacrine cell dendritic branches in stratum 5 appear later at P15 and become pronounced only by P20 [46]. This thus insinuates that contrary to the presence of Cx36 in many retinal microcircuits, its expression maximum is closely linked to the formation of retinal night vision pathways. Although most studies followed the Cx36 expression changes during the early postnatal (P1–P20) and adult (P30–P60) life, little is known about the aging (>P300) retina (note that all the postnatal timepoints here refer to the murine retina as the most popular experimental models of retinal research). Based on our own unpublished observations, the aging mouse retina seems to lose much of the Cx36 immunoreactivity in terms of the number of antiserum-stained plaques in the IPL. However, it is unclear if this is related to the loss of Cx36 GJs and the corresponding function, or if it is due to other issues including: (i) Cx36 being replaced by other Cx subtypes in the aging retina; (ii) the a-Cx36 antiserum not recognizing the protein that might change the active conformation with time; (iii) the Cx36 GJs reducing in size over time and most of them reducing beyond the detection level or (iv) the aging tissue being more sensitive to the experimental manipulations and samples start disintegrating when the same experimental protocol is utilized. Thus, future work should clarify this question. In contrast to the monotonous increase in Cx36 throughout postnatal development, the expression profile of Cx45, the other major subunit utilized by retinal neurons, displays a continuous decrement from P1 to adulthood [58]. This correlates with the increasing downregulation of the Cx45 mRNA level during development in the rat retina [45,59] (our own unpublished observations, Figure 3a,b). This differential expression pattern of these two Cx subunits (Cx36 and Cx45) could be explained by the replacement or substitution of some of the Cx45 GJs by Cx36 connections during the early postnatal life.

Besides the above two prevalent Cx subunits, Cx30.2 is also expressed in some RGC connections [43], whereas Cx50 and Cx57 subunits form horizontal cell GJs [29,61,62]. The developmental changes of these latter Cx subunits have not been established yet; although, our unpublished observations indicate that the Cx57 mRNA transcript level is monotonously increasing during postnatal development P0-P15 and displays a plateau in the adult tissue (unpublished observations, Figure 3c). Since this latter Cx is specifically expressed by retinal HCs, its expression level likely follows the maturation of the outer retinal wiring especially those formed by horizontal cells. One other well-studied subunit is Cx43 whose expression, similar to those of Cx36, displayed an increased expression from early postnatal to adult retinal stages [58]. Cx43 is almost absent in retinal progenitor cells at P1, but it becomes prominent at P10. Much of this label at this stage seems to coincide with vimentin-positive Muller cell processes. In addition to protein expression, quantitative PCR showed that Cx43 and vimentin transcripts also shared very similar temporal expression patterns indicating the strong correlation between Muller cell maturation and Cx43 functioning. Besides Muller cells, Cx43 expression was also prevalent in the nerve ganglion cell layers later at P5–P15 [58]. At least some of this latter staining can be accounted for by the Cx43 expression of astrocytes, based on the colocalization of Cx43 with GFAP from P15 [45]. The importance of Cx43 is also highlighted by the expression profile alterations in different diseases like in ischemia [63] and in diabetic retinopathy in the retinal vasculature [64,65] In addition to their roles in the adult retina, some GJs have been suggested to play a role in neuronal development as well. It has been shown, using neuronal precursor cells, that Cx43 expression levels increased with postnatal age in the subventricular zone. This suggested that Cx43 might be a negative regulator of cell proliferation, even though it promotes proliferation during embryonic development [66]. In addition, Cx45 has been hypothesized to enhance synaptogenesis based on its frequent colocalization with Synaptophysin and the elevated number of heterotypic Cx45/Cx36 gap junctions prior to eye opening [67].

## 3. Light Adaptation-Induced Changes in Connexin Expression Levels

During the daily darkness/brightness cycle, the retina operates over a ~50-billion-fold change in illumination and maintains a broad dynamic range. This involves a tremendous capacity of light adaptation including several different mechanisms. One of these involves a shift of signaling between the rod and cone photoreceptor-initiated pathways. This rerouting of visual signals within the retina happens slowly during dusk and dawn but also occurs more quickly when one enters a dark room during daylight (or exits from a dark room to a bright exterior). For the smooth transition between signaling systems, the retina utilizes an intermediary route, called the secondary rod pathway, which relies on Cx36 GJs between rods and cones. The Cx protein subunit turnover takes only a few hours, leaving an effective time window for a potential light adaptation-induced Cx transcript and protein expression change [68]. However, the sudden light condition changes require considerably faster mechanisms than the expression of new channel-forming Cx subunits to increase GJ conductance. Therefore, rather than offsetting the Cx biosynthesis/degradation balance, the opening probability of GJs increases (GJs open) in darkness and dim light and decreases (GJs close) in bright light. When open, rod/cone GJs transmit rod signals through cones to cone bipolar cells that eventually target RGCs. The key step in this process is the alteration of the GJ opening probability, which is performed by the gradual phosphorylation/dephosphorylation of the Cx36 protein subunits. When Cx36 subunits are phosphorylated, the Cx36 GJ conductance between cones and rods increases [69]. In daylight, activation of dopamine D2/D4 receptors in cone photoreceptors inhibits adenylate cyclase activity and prevents the cAMP-dependent protein kinase A (PKA)-mediated Cx36 phosphorylation. Therefore, Cx36 channels are kept dephosphorylated and closed during bright light conditions. During darkness, on the other hand, the inhibition of adenylate cyclase is relieved and phosphorylation of Cx36 results in an increase in GJ conductance. In addition to dopamine, whose release is induced by light, intracellular adenosine levels also play a role in regulating photoreceptor GJ coupling. When adenosine levels are high at night, adenosine opposes the dopamine effect and enhances coupling, whereas it reinforces the dopamine effect during the day when both compounds suppress coupling [70]. Adaptation to scotopic (rod-mediated) vision, besides changes in the photoreceptor GJ gating, occurs in the rest of the retina as well. The electrical circuitry in the retina for example is dynamically regulated by light, suggesting that it plays a major role in adaptation [11,71,72,73,74,75,76,77,78]. For example, adaptational changes have been shown to modulate the extent of both tracer and electrical coupling of HCs that are maximized under dim ambient light conditions and diminish as the retina is dark- or light-adapted from this level [77,79,80,81,82,83,84]. Light adaptation increases expression of Cx57 [85] allowing for the slow adaptation of circuits to changing light conditions. A faster mechanism through mediators like dopamine or NO, both of which are released in the retina on a light-dependent matter, also affects signaling across the extensively coupled HC array. Dopamine, for example, decreases the HC GJ conductance acting through the intracellular messenger cAMP, thereby producing a concomitant reduction in receptive field size [86,87,88,88]. Through a separate intracellular cascade, NO also alters intracellular levels of cGMP and increases HC electrical coupling [89,90]. Similarly, under dark-adapted conditions, AII ACs in the proximal retina are coupled in relatively small groups, but the size of coupled AII cell groups dramatically increases following exposure to dim background light. This then reverts and AII cells uncouple secondarily when the retina is further adapted to even more intense background light [72]. AII cells form homologous GJs with nearby AIIs using Cx36 subunits [31,32], whereas their heterologous GJ contacts with ON cone bipolar cells comprises a mixture of Cx36 and Cx45 [31,32,34,91,92]. These differences in the subunit composition of the two sets of AII GJ populations can explain their different conductance and pharmacology. While homologous AII GJs are modulated by levels of the intracellular cAMP, those connecting AIIs to ON cone bipolar cells are modulated by cGMP [11]. Although, many other retinal interneurons (BCs and ACs) display GJ connections with their neighbors [15,93], the light-dependent changes in their conductance and/or Cx expression levels have not been studied yet. Besides GJs in PRs and the retinal interneurons, the light-modulated GJ coupling of the retinal output neuron RGCs has been studied for some of the subtypes. RGCs were found to be extensively coupled in the inner retina and light adaptation induces a monotonous expansion of coupled RGC arrays [43,94,95]. It has been proven for example that among ON–OFF direction-selective ganglion cells (DSGCs), superior preferring cells exhibited a broadening in their direction tuning at low light levels; however, the responsible intracellular mechanisms are not known [96]. On the other hand, it is well documented that light adaptation activated intracellular cascades that eventually lead to PKA-induced phosphorylation of Cx36 in OFF α-GCs arrays, whereas light adaptation initiated another intracellular cascade in the G1 RGC coupled network where a PKC-mediated phosphorylation of Cx30.2 occurred [43,97,98,99]. Less is known about the light adaptation-induced changes in the levels of Cx mRNA transcripts and corresponding proteins. It has been shown that Cx57, Cx45, Cx36 and Cx43 mRNA transcript levels were repressed by prolonged darkness (occurring between 3 h and 7 days). These changes were accompanied by a similar dark-induced downregulation of Cx36 and Cx43 protein subunits [45,58]. These results, therefore, indicate that the time window provided by the timescale of illumination changes during the night/day cycle could be effectively utilized to modulate the Cx transcript and protein expression changes [68] and corresponding GJ functioning in the retina.

## 4. Circadian Rhythmicity and GJ Function

Previous sections described how retinal GJs and their Cx building blocks endure expression changes throughout the lifespan of an organism and summarized how quick environmental changes affect the intracellular milieu and corresponding GJ gating in various retinal cells. The responses of the retinal tissue to changes on these two timescales overlap considerably. For example, adaptation to altered light conditions initially involves only the on- or offset of intracellular signaling mechanisms and the corresponding change in GJ gating. However, as pointed out in the previous section, longer lasting light adaptation eventually exerts Cx expression changes as well [45,58]. Thus, the daily cyclic alterations of the environment occur on overlapping timescales that evoke both quick molecular responses and longer-lasting mRNA and protein expression changes in neurons of the retina. That said, the daily light–dark cycle regulates neuronal activity in the retina, and a corresponding switch in signaling through rod- and cone-specific retinal circuitry occurs (see above). In addition to the rerouting of retinal signals, the expression levels of melatonin and dopamine [100] are also in the counter phase [101], serving as reciprocally inhibitory messengers for darkness and light, respectively [95,102]. These initial light/dark-induced changes in melatonin and dopamine levels mediate the expression of a cohort of other molecules, including the upregulation of protein kinase C in BCs in daylight conditions and the upregulation of parvalbumin in AII ACs in dark conditions [103,104]. Many of these changes can be explained by the diurnal rhythm and adaptation to ambient light levels (see above). However, maintained rhythmicity of melatonin synthesis has been observed in retinal cultures even in constant darkness [102,105,106]. Clearly, mechanisms under the control of the circadian clock also act on this overlapping time scale and thus contribute to some of the previously presented quick molecular changes and induce alterations in mRNA and protein expression with a slower onset. In this context, “diurnal rhythm” refers to responses to environmental cues (e.g., changes in the levels of background light), whereas a circadian rhythm is instead controlled by an endogenous oscillator. Besides melatonin, expression of certain retinal proteins, including iodopsin in cones [107], parvalbumin in AII ACs [104] and tryptophan hydroxylase in ACs [108], have been reported to follow a rhythm independent of the light-induced diurnal cycle. Moreover, many genes in the retina follow circadian rhythmicity, including core circadian clock genes (Per1, Per2, Cry1, Cry2 and Bmal1; [109], transcriptional activators Rora, Rorb and Rorc; 51; [110], as well as miRNAs miR-103, miR-106b, miR-124a, miR-182, miR-422a and miR-422b51), which modulate the translation of mRNA transcripts into functional proteins. This converging evidence indicates that the retina has a biological circadian clock and that the switch from rod to cone vision involves the diurnal and also the circadian rhythm. The outer nuclear, inner nuclear and ganglion cell layers of the retina have been found to each contain an independent layer-specific circadian oscillator (93). It has been demonstrated for example that the relatively weak rod–cone electrical coupling in the retina is under circadian regulation [111,112]. It is known that during the day, the decreased release of melatonin results in increased dopamine release from dopaminergic ACs [111,112]. Dopamine then activates D2-like receptors on rods and cones, which results in decreased intracellular cAMP levels, decreased PKA activation and decreased Cx36 phosphorylation in the PRs [113,114,115,116,117,118,119,120,121]. Decreased Cx36 phosphorylation is correlated with decreased rod–cone GJ conductance [122]. In contrast, during the night, the retinal clock increases melatonin synthesis, which reduces extracellular dopamine levels and D2-like receptor activation, resulting in increased Cx36 phosphorylation [112,117]. Increased Cx36 phosphorylation is correlated with increased rod–cone GJ coupling [122,123], allowing the convergence of rod signals into cone pathways [69]. Whether the circadian rhythm regulates Cx36 mRNA transcript and protein levels in addition to Cx36 phosphorylation seems to be a controversial question. One study reported higher levels of Cx36 mRNA transcript and protein during the subjective night than during the day [124], but other studies have not found such circadian regulation [99,123]. Interestingly, in addition to the increased rod–cone GJ coupling during the night, it has been found that rod–rod GJ coupling is also under circadian regulation in the mouse retina, with more coupling during the night than during the day [125,126]. This trend is observed in the melatonin-proficient CBA/Ca mouse strain but not in the melatonin-deficient C57BL/6 mouse, indicating that the circadian regulation of rod–rod GJ coupling is also dependent on the melatonin/dopamine neuromodulator interaction [126]. In contrast to PR coupling, gating of other retinal GJs may not be under the control of the circadian clock. It appears for example that activation of D1 receptors, which causes GJ uncoupling between HCs, is not under circadian regulation as the extent of GJ coupling between horizontal cells has been found to be equal during night and during the day in the fish retina [127]. Independent of the melatonin/dopamine interaction, extracellular levels of adenosine are also under circadian regulation, such that extracellular adenosine levels are high during the night and lower during the day [112,128]. It has been demonstrated in the goldfish retina that the activation of adenosine A2A receptors on PRs during the night also contributes to increased rod–cone GJ coupling, thereby further enhancing the difference in rod–cone GJ coupling between night and day [128]. Thus, adenosine and dopamine both play a role in the circadian variation of rod–cone GJ coupling, but they do so with opposite effects on the PKA pathway and on Cx36 phosphorylation [122,128]. In addition to the D2-like and A2A receptors, the cannabinoid CB1 receptors on PRs have also been found to play a role in enhancing the difference in rod–cone GJ coupling between night and day. During the day, when the D2-like receptors are activated, endogenous activation of CB1 receptors decreases rod–cone GJ coupling. During the night, when the D2-like receptors are not activated, CB1 receptor activation increases rod–cone GJ coupling. Thus, it appears that the effect of CB1 activation on the extent of rod–cone GJ coupling depends on the activation state of the D2-like receptors [129]. The above cases exemplify that the GJ function is strongly modulated by not only the level of background light but also by the status of the circadian biological clock. Besides a few well-studied GJ connections (those connecting PRs or those between AII ACs), however, there is little or no information on similar daily changes in the rest of the electrical synapses in the retina. It is also unknown if the above-indicated molecular changes mediated by both diurnal and circadian regulators are accompanied by Cx transcriptional modifications. According to the scarce literature [122,123,124] and our own unpublished observations, such Cx expression responses do exist [46,60]. In the rat retina, we observed daily fluctuations of the Cx36 mRNA levels that reached the maximum both in the middle of the day and the middle of the night, while it was downregulated during dusk and dawn (Figure 4).

Although the observed Cx36 mRNA level changes were not robust (and were not significant), they may correspond to the increasing GJ gating and Cx36 protein need in the coming dusk and dawn periods (note that mRNA expressions should precede protein level elevations thus mRNA transcript levels peak hours prior to when protein expression needs a boost). It is unknown, however, how and to what extent external diurnal and internal circadian factors contribute to these changes. Future work will enlighten these details along with those describing how other retinal Cx transcript and protein levels are controlled by the daily rhythm and if circadian mechanisms partake in that process. It has also been demonstrated that when the retina is treated with a GJ blocker, the circadian period of each layer-specific oscillator increases significantly compared to the normal circadian period of the whole retina (93). This thus reflects that GJs and Cx building blocks are not just under the control of daily cyclic changes but also partake in the functioning of the circadian clock. This indicates that an important mutual interaction exists between GJ coupling and the retinal circadian rhythm.

## 5. Conclusions

In this present review, we summarized latest advances in the field of GJ signaling in the mammalian retina with a focus on the expression alterations of GJ-forming retinal Cx subunits as a response to intrinsic and environmental changes. Such changes include the cyclic modulation of the intracellular molecular milieu driven by the circadian clock as well as cyclic diurnal and monotonous alterations throughout the postnatal life of an organism. While the Cx mRNA and protein expression alterations appear more prominent, changes occur on a longer timescale (throughout postnatal life or extended light/dark adaptation) and they also seem to follow quick changes of the internal and external environment, including circadian rhythmicity and diurnal light level alterations, respectively. By demonstrating these changes in GJ signaling as well as expression levels of the Cx building blocks, this work exemplifies how signaling through a sensory organ adapts to the altered environmental conditions. In addition, this converging evidence also pinpoints the importance of the standardization of experimental conditions, such as the age of the animal models, the timing of experiments and also the light conditions.

## Figures and Tables

**Figure 1 biomolecules-13-01119-f001:**
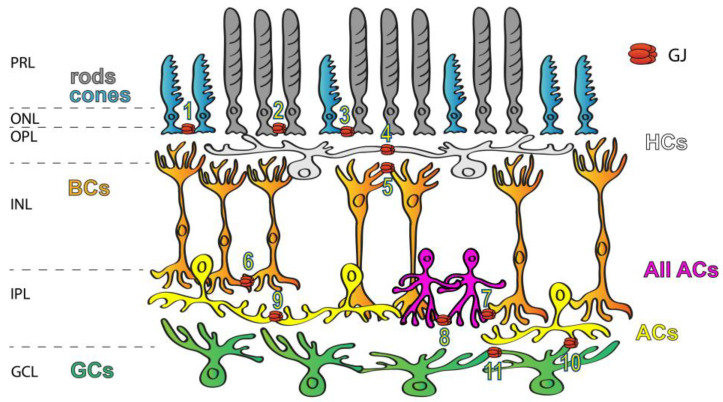
Expression of GJs in the mammalian retina. GJs connecting retinal neurons are ubiquitous and they form a variety of connections, including rod/rod (1), cone/cone (2), rod/cone (3), HC/HC (4), dendritic BC/BC (5), axonal BC/BC (6), AII AC/ON BC (7), AII AC/AII AC (8), AC/AC (9), AC/RGC (10) and RGC/RGC (11) GJs. Labels to the left mark the layers of the retinal tissue: PRL—photoreceptors layer, ONL—outer nuclear layer, OPL—outer plexiform layer, INL—inner nuclear layer, IPL—inner plexiform layer and GCL—ganglion cells layer.

**Figure 2 biomolecules-13-01119-f002:**
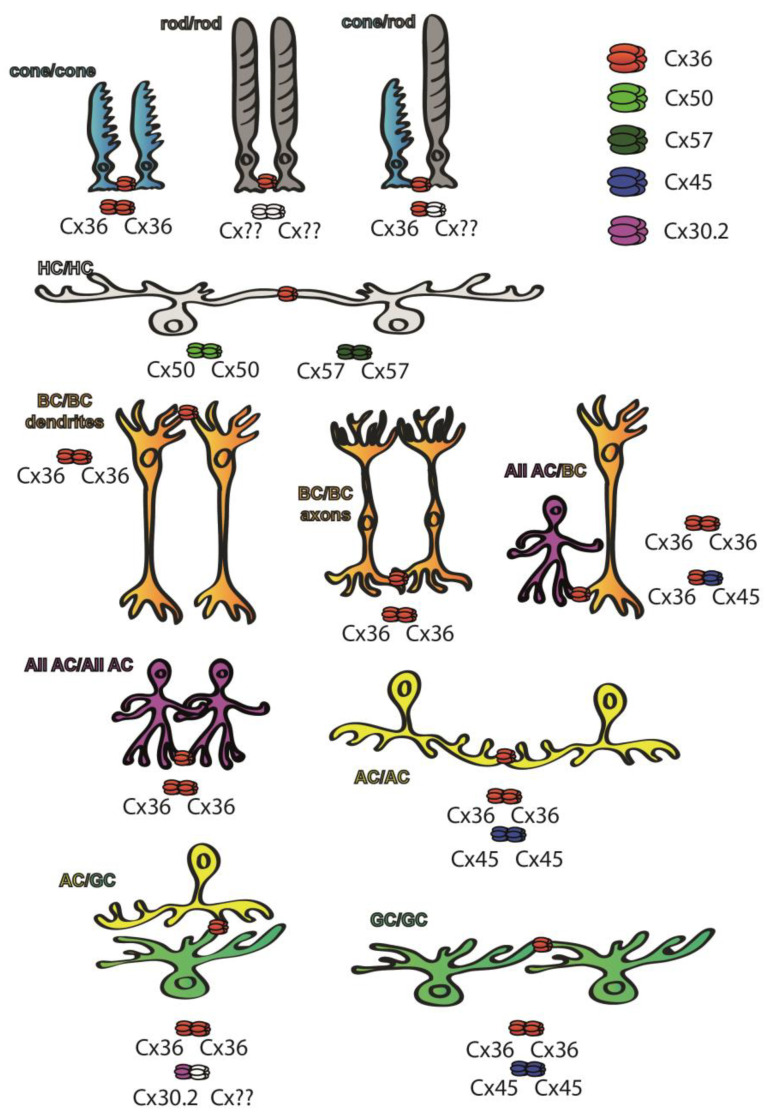
Cx subunit composition of GJs connecting neurons of the mouse retina. While Cx36 subunits are ubiquitous throughout the retina [27], Cx50 and Cx57 subunits are only found in outer retinal HC GJ connections [29,30], whereas Cx45 is majorly expressed in the inner retina [28]. Some of the retinal GJs are homotypic thus connecting neurons that express the same Cx subunits in their hemichannels, including cone/cone GJs [27,31,32,33,34,35,36], AII GJs [31,32,33] that comprise Cx36 or ON–OFF DS RGCs that express Cx45 [41]. Other connections are clearly heterotypic, thus connecting neurons expressing at least two different Cx subunits. The best known such connection is formed between BC axon terminals and AII ACs, in which AII cells express Cx36 while BCs express either Cx36 or Cx45 depending on their cellular subtype [37,38]. A third cohort of connections comprise a connexon whose identity has been confirmed, while the connecting hemichannel has not been determined yet (marked by ??). These latter connections include rod hemichannels [27,31,32,33,34,35,36], GJs that connect α-GCs into an array [39,40,42] or the ACs that form GJs with Cx30.2-expressing A1 type RGCs [43].

**Figure 3 biomolecules-13-01119-f003:**
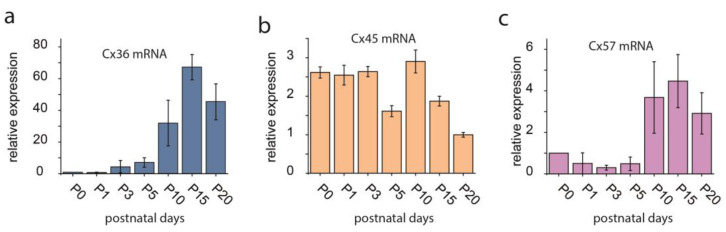
Expression of the Cx36 (**a**), Cx45 (**b**) and Cx57 (**c**) mRNA transcript in the rat retina throughout the early postnatal life (P0–P20). Adopted and modified from Kovács-Öller et al., 2014 and Kovács-Öller PhD thesis [46,60]. Error bars represent SD for data from n = 3 independent samples.

**Figure 4 biomolecules-13-01119-f004:**
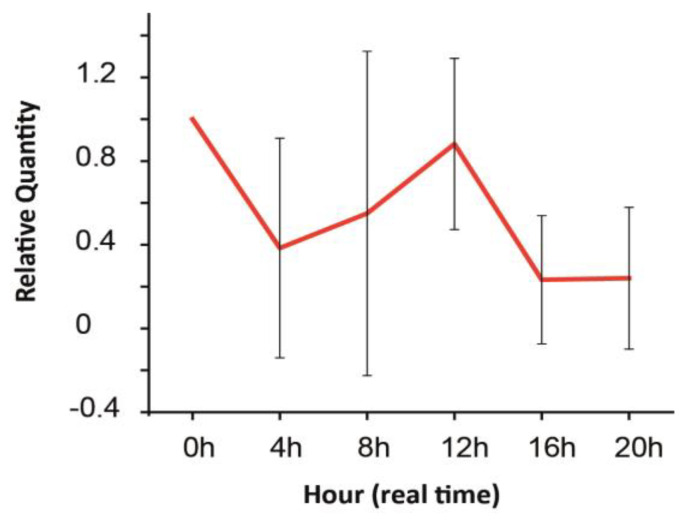
Expression of the Cx36 mRNA transcript in the rat retina throughout an entire 24 h day period. Adopted and modified from Kovács-Öller et al., 2014 and Kovács-Öller PhD thesis [46,60]. Error bars represent SD for data from n = 3 independent samples.

## Data Availability

Not applicable.

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
