# Peer review of "Extrinsic and Intrinsic Factors Determine Expression Levels of Gap Junction-Forming Connexins in the Mammalian Retina"

_biomolecules, 2023, doi:10.3390/biom13071119_

Round 1

Reviewer 1 Report

This is an interesting, well-written review article that describes how gap junctions are regulated in the retina by the cyclic modulation of the intracellular molecular milieu driven by the circadian clock as well as by light/dark cycles and postnatal development

Specific comments:

Abstract: I would reorganize the abstract to emphasize the main focus of the review which is changes in gap junctional coupling and expression in the healthy retina. The first sentence of the abstract could read “Gap junctions (GJs) are not static bridges.”   Instead, they are constantly undergoing adaptive changes in the healthy retina in response to ….

1.1 GJ and Connexin Building Blocks

Line 51: omit “also”

Line 54: rephrase to read “ hemichannels located in plasma membranes of neighboring cells.”

Line 58: add “molecular permeability and unitary conductance of each pore.”

Line 58: new sentence “ The functional properties of the pore can also be modified by the intracellular molecular milieu….”

1.3 Connexin Composition of Retinal GJs

It would be helpful to add a figure showing the different layers of the retina.

Lines 96-98: The role of Cx45 in mediating signals of the night vision pathway should be more clearly described.

Lines 110-111: Many of the rapid changes in gap junctional coupling appear to be due to changes in post-translational modification (i.e. phosphorylation) rather than changes in protein expression.

2. Postnatal changes in connexin expression

Line 128-129. Puncta is pleural

Figure 1 and 2. What do the error bars represent?

Figure 2.  Are the observed changes statistically significant?

The paper was generally well written.  However, some minor editing of the English language is required.  See previous comments

Author Response

Referee#1

We thank the referee for his/her work and for the critics and comments that helped us to make this study significantly better and scientifically sound.

  1. Abstract: I would reorganize the abstract to emphasize the main focus of the review which is changes in gap junctional coupling and expression in the healthy retina. The first sentence of the abstract could read “Gap junctions (GJs) are not static bridges.”   Instead, they are constantly undergoing adaptive changes in the healthy retina in response to ….

We rephrased this sentence according to the suggestion.

  1. Line 51: omit “also”

„also” has been omitted.

  1. Line 54: rephrase to read “ hemichannels located in plasma membranes of neighboring cells.”

We rephrased this sentence accordingly.

  1. Line 58: add “molecular permeability and unitary conductance of each pore.” Line 58: new sentence “ The functional properties of the pore can also be modified by the intracellular molecular milieu….”

We added the line and the sentence was corrected according to the suggestion.

  1. It would be helpful to add a figure showing the different layers of the retina.

Referee#2 also requested a figure to show the GJ sites of the mammalian retina, thus we created a schematic (new Figure 1) that shows GJs as well as the retinal layers.

  1. Lines 96-98: The role of Cx45 in mediating signals of the night vision pathway should be more clearly described.

We rewrote these sentences providing extra information of the Cx45 contacts between AII ACs and ON cone BCs.

  1. Lines 110-111: Many of the rapid changes in gap junctional coupling appear to be due to changes in post-translational modification (i.e. phosphorylation) rather than changes in protein expression.

We added the sentence to the text and also modified other sentences of the same paragraph to place it in a context.

  1. Line 128-129. Puncta is pleural

We corrected the rest of the sentence to match the plural word (puncta).

  1. Figure 1 and 2. What do the error bars represent?

The measurements were repeated in n=3 animals for each postnatal ages (new Figure 3) and each circadian time point (new Figure 4). The error bars represent the SD on the average of the results. We supplemented the figure legends with this detail.

  1. Figure 2.  Are the observed changes statistically significant?

We state it the main text these results were not statistically significant (this is one of the reasons why the data have only been published in the first author’s PhD dissertation). However, the curves still show a trend of the expressional changes that suggests such circadian regulation.

  1. The paper was generally well written. 

Thank you for the positive comment!

Reviewer 2 Report

The stated goal of this review is to summarize the plasticity of gap juncito0ns in the retina with respect to three distinct temporal regimes: rapid, diurnal, and long term.  Such a clear review, including how each cell type and connexin is manipulated by extrinsic and intrinsic factors, would be of high interest to the field.  The present review presents only two figures, one showing ontogeny of mRNA expression for three connexins and the other diurnal fluctuations in Cx36 with vast variance, both of which are rather minor aspects of the review's emphasis.  It thus misses opportunities to summarize distinct cell types, connexins and time regimes with clear figures illustrating how circuits are modified and what that means for visual processing in the retina.  

As written, the text is disappointingly dense, repetitive and confusing.  WIth clear figures, this could become a high impact and timely review that disentangles the dopamine, adenosine and cannabinoid modulation and points out how modulation impacts visual functions.

Two sections that do not fit well with the rest of the document regard miRNAs, which are not clearly linked to connexins, and hemichannels, a topic that is very confusingly presented.

A few specific comments;

l18 and ff.  Expressional is incorrectly used

Abstract emphasizes difference between connexin flexibility and tissue deterioration, a distinction that seems strange.

l60.  what does best known mouse and human genomes mean?

l71.  What is meant here?  COnnexin isologs/homologs exist across the entire vertebrate family. l77.  WHt does hgihly conductive mean?  THese are very low conductance connexins.

l90.  Please illustrate connexin distriubtion in retinal cells.

l119.  Invertebrate gap juncitons are formed by innexins, not connexins.

l131.  Prtein abundance precedes mRNA?

l143. plaques

redundancy throughout

Author Response

Referee#2

We thank the referee for his/her critics and comments. These all helped us to make this study significantly better, especially the recommendations for the extra figures.

  1. The present review presents only two figures, one showing ontogeny of mRNA expression for three connexins and the other diurnal fluctuations in Cx36 with vast variance, both of which are rather minor aspects of the review's emphasis. It thus misses opportunities to summarize distinct cell types, connexins and time regimes with clear figures illustrating how circuits are modified and what that means for visual processing in the retina.  

We added two additional figures (new Figures 1 and 2). Figure 1 is a schematic that summarizes retinal cell types and layers and shows the most relevant GJ connections formed by various retinal cell types. Figure 2 is also a schematic that summarizes the Cx makeup of various retinal GJ connections (note that the original two figures now became Figures 3 and 4).

2/ As written, the text is disappointingly dense, repetitive and confusing.  WIth clear figures, this could become a high impact and timely review that disentangles the dopamine, adenosine and cannabinoid modulation and points out how modulation impacts visual functions.

As pointed out above, we added two additional explanatory figures (new Figures 1 and 2). We hope that these figures help to absolve much of the textual information and thus clarify this review. Since the point of this review was to show changes of Cx mRNA and protein expressional patterns we only mentioned intracellular molecular mechanisms that are regulated by the environmental and/or intrinsic factors and may or may not affect the Cx expressional patterns. While dopamine, adenosine and cannabinoid driven mechanisms initiated as a response to the changing environment the direct connection between the corresponding intracellular cascades and the regulation of Cx mRNA transcription and/or protein translation is yet unclear. 

  1. Two sections that do not fit well with the rest of the document regard miRNAs, which are not clearly linked to connexins, and hemichannels, a topic that is very confusingly presented.

As suggested by this referee we deleted sections that described miRNA expressional changes and their potential involvement in Cx protein expression. By deleting these sections this review also became shorter and more succinct thus addressing one of the previous criticisms as well.

  1. 4. l18 and ff.  Expressional is incorrectly used

We replaced the word ’expressional’ to ’expression’ throughout the text.

  1. Abstract emphasizes difference between connexin flexibility and tissue deterioration, a distinction that seems strange.

Cx (and related GJ) mRNA and protein expression alterations occur as a response to degenerations or external insults of the retinal tissue as well as to accommodate for the healthy tissue to natural changes in the environment. When tissue deterioration is in the focus it is crucial to delineate the extent natural causes may contribute to Cx expression changes in order to evaluate the severity of pathological changes to the degenerating tissue.

  1. l60.  what does best known mouse and human genomes mean?

We admit that this was negligent wording and now we rephrased this sentence in the revision.

  1. l71.  What is meant here?  COnnexin isologs/homologs exist across the entire vertebrate family.

Yes, exactly. Just to provide one example; mammalian GJs formed by Cx36 subunits have Cx35 (skate, perch, zebrafish) and Cx34.5 (perch) homologs in the fish retina.

  1. l77.  WHt does hgihly conductive mean?  THese are very low conductance connexins.

The transmembrane conductance of mouse Cx57 and Cx50 GJs formed between HCs are in the 0.8-3 nS and the 2.2 – 5 mS ranges, respectively. These values are several magnitudes higher than those measured for low conductance Cx36 GJs (connecting PRs, RGCs or ACs) that are all in the pS range.

  1. l90.  Please illustrate connexin distribution in retinal cells.

We provided a schematic in Figure 2 of the revision to show the Cx distribution for neuronal GJ connections in the mammalian retina.

  1. l119.  Invertebrate gap juncitons are formed by innexins, not connexins.

Yes, but we did not state that in the text. The section goes like this: Electrical synapses (GJs) have been found in both invertebrates including the sea urchin, starfish, ascidians, and various species from all vertebrate classes (48). Since those early days, it has repeatedly been shown that the expression of GJ forming Cx transcripts displays a temporal pattern during organogenesis both outside the brain (49–51) and in many brain areas (52–56). Thus, we already talk about vertebrates when we first mention Cx transcripts and proteins. To avoid confusion, we modified this section to be more precise.

l131.  Prtein abundance precedes mRNA?

The section goes like: It appears that Cx36 immunoreactive puncta appears first in a few cells in the presumptive ganglion cell layer at P1 (43) and becomes abundant in the two plexiform layers at P10 (44). This latter developmental stage also coincides with a peculiar surge in Cx36 protein expression in WB measurements and just precedes the maximum of Cx36 mRNA expression at P15 (43, 44). Thus, the P10 developmental stage is the one that precedes the Cx36 mRNA maximum (not the Cx36 protein expression).

l143. Plaques

Yes, that was a typo and we corrected that.

Round 2

Reviewer 2 Report

The authors have added two excellent figures and removed a somewhat distracting section on miRNAs.  This is now a highly readable, informative reveal of the state of gap junctions in the retina.

I have a few minor suggestions that may further improve the manuscript: '

It would be helpful if font size of numbers indicating the gap junctions could be increased in Fig 1, and readers would likely also benefit from cell types being named to the left of Fig 2.

Yes, there is a range of unitary conductance of GJ channels (Cx36 ~30 pS), but the largest is about 300 pS, not the high conductances you mention in rebuttal.

Yes, your statement involving invertebrate GJs, called them electrical synapses (which is formally incorrect, since they are not between neurons), so innexin was not necessary (but synapse might be changed).  

Author Response

Referee#2 2nd revision

Again, we thank to referee#2 for his/her work (suggestions and critics)! As we point it out below we made all the required changes in both the text and the Figures.

The authors have added two excellent figures and removed a somewhat distracting section on miRNAs.  This is now a highly readable, informative reveal of the state of gap junctions in the retina.

Thank you for the credits!

It would be helpful if font size of numbers indicating the gap junctions could be increased in Fig 1, and readers would likely also benefit from cell types being named to the left of Fig 2.

We agree with the referee opinion, the numbers were too small in Figure 1. We increased the size of the fonts and also gave them a color to better prevail against the background. We also added the cell types to Figure 2.

Yes, there is a range of unitary conductance of GJ channels (Cx36 ~30 pS), but the largest is about 300 pS, not the high conductances you mention in rebuttal.

Well, the high conductance stands for the HC GJs formed by either Cx50 or Cx57 subunits. We agree that the Cx36 GJs possess relative low conductance.

Yes, your statement involving invertebrate GJs, called them electrical synapses (which is formally incorrect, since they are not between neurons), so innexin was not necessary (but synapse might be changed).  

The referee is right, we admit that the term was erroneously used originally. We corrected the second revision accordingly.